# Generalized Linear Integer Numeric Planning

## Abstract

Classical planning aims to find a sequence of actions that guarantees goal achievement from an initial state. The representative framework of classical planning is propositional logic. Due to the weak expressiveness of propositional logic, many interesting real-world applications cannot be described as a classical planning problem. Some extensions such as numeric planning and generalized planning are therefore proposed. In this paper, we focus on a generalized version of numeric planning, namely generalized linear integer numeric planning (GLINP), requiring each numeric variable to be an integer, and initial states to be formalized as a numeric formula that represents possibly infinitely many states. GLINP is a more expressive planning formalization than qualitative numeric planning. In addition, we develop an approach to synthesize solutions to GLINP problems. This approach generates a solution which can satisfy all instances of the domain as long as the set of initial states is representative. Finally, we evaluate our approach on several benchmarks, and experimental results demonstrate the effectiveness and scalability of our proposed approach.

## Introduction

Along the AI history, the planning community has focused on classical planning that identifies a sequence of actions that guarantees goal achievement from an initial state. The representative framework of classical planning is propositional logic. Due to the weak expressiveness of propositional logic, many interesting real-world applications cannot be described as a classical planning problem. Therefore, some extensions to classical planning are proposed. One extension is numeric planning (Do and Kambhampati 2001; Haslum and Geffner 2001), which involves not only propositional variables but also numeric variables. Another extension is generalized planning (Levesque 2005; Srivastava, Immerman, and Zilberstein 2011), which solves planning problems for possibly infinitely many initial states rather than a single state. However the above two extensions are in general undecidable (Helmert 2002; Levesque 2005).

Srivastava et al. (2011b) proposed a decidable class of numeric and generalized extensions to classical planning, namely qualitative numeric planning (QNP). There are,

however, two restrictions imposed on numeric planning: (1) any formula describing the initial states and the goal states is a Boolean combination of literals of the forms of $v > 0$ or $v = 0$; and (2) the effects of actions decrease or increase the value of some variables by an unspecified amount. Under these two restrictions, the state space of a QNP problem can be compressed into a finite one with size $2^{|V|}$ where $|V|$ is the number of numeric variables, and thus QNP is decidable.

However, some integer numeric planning domains still cannot be modeled by QNP due to its two restrictions. Consider the following example, which is a modified form of the sailing problem (Scala, Haslum, and Thiébaux 2016).

**Example 1** (Sailing)**.** There are a sailing boat and people to be rescued on an ocean with an unbounded area. The task of the sailing boat is to reach the specific area, in which the people is, so as to rescue them. The positions of the boat and the people are formalized by their coordinates $(x, y) \in \mathcal{Z}^2$. Initially, the boat is in a position $(x, y)$ satisfying $x + y \geq 1$ and the people are in the area where the position satisfies the linear inequality $(x + y \geq -2) \wedge (x + y \leq 0)$. Each of the $8$ possible movements of the boat is formalized in the geometrical space defined by $\mathcal{Z}^2$, for example, the action $d$ moves to the north decreasing the $y$-coordinate of the boat by $2$.

It can be easily observed that (1) the formulas representing initial states and goal states violate the first restriction, and (2) the effects of actions are too accurate to comply with the second restriction. Hence, QNP is not a suitable formalization for the sailing problem.

In order to address the deficit, we propose a generalized version of numeric planning, namely generalized linear integer numeric planning (GLINP). Compared to numeric planning, GLINP requires that all numeric variables are integer ones, and the initial states are represented by a linear numeric formula that is able to capture possibly infinitely many states. When considering numeric planning with integer numeric variable, GLINP is a more expressive planning formalization than QNP in integer numeric planning. To capture the notion of solutions to GLINP, we introduce a more general solution, namely planning programs, consisting of an empty plan, primitive actions, sequential, branch and loop structures. In addition, we develop a generation approach to synthesize planning programs. The generation stage firstly infers a skeleton of planning programs, and

then completes the conditions of branch and loop structures. Finally, we evaluate our approach on several benchmarks, and experimental results demonstrate the effectiveness and scalability of the proposed approach.

## Preliminaries

In this section, we first introduce the concepts of integer arithmetic with first-order logic ($\mathrm{IA^{FO}}$), along with its well-known decidable fragment of linear integer arithmetic with propositional logic ($\mathrm{LIA^P}$) and regular expressions.

Let $\mathcal{Z}$ the set of integers, $\mathcal{N}$ the set of natural numbers, $\mathtt{Var}$ a set of variable symbols, $\mathcal{F}$ a set of function symbols and $\mathtt{Pred}$ a set of predicate symbols. The sets of *numeric terms* ($\mathtt{Term}$), *atomic formulas* ($\mathtt{Atom}$) and *formulas* ($\mathtt{Form}$) for $\mathrm{IA^{FO}}$ is defined by the following grammar:

$$e \in \mathtt{Term} :: c \mid x \mid e + e \mid e \times e \mid F(e, \cdots, e)$$

$$\alpha \in \mathtt{Atom} :: e = e \mid e < e$$

$$\phi \in \mathtt{Form} :: \alpha \mid \neg\phi \mid \phi \wedge \phi \mid \exists x \phi$$

where $c \in \mathcal{Z}$, $x \in \mathtt{Var}$, $v \in \mathcal{F}$.

The formula $\phi_1 \vee \phi_2$ is the shorthand for $\neg(\neg\phi_1 \wedge \neg\phi_2)$ and $\forall x \phi$ is the shorthand for $\neg\exists x(\neg\phi)$.

A formula $\phi$ or a term $e$ is *closed* if it contains no free occurrence of a variable. For an interpretation $M$ of $\mathrm{IA^{FO}}$, it is required that its domain is the set $\mathcal{Z}$ of integers, each $m$-ary function symbol is interpreted as an $m$-ary integer function $f(M) : \mathcal{Z}^m \to \mathcal{Z}$, and each $m$-ary predicate symbol is interpreted as an $m$-ary integer relation $p(M) \subseteq \mathcal{Z}^m$. The addition and multiplication functions, and the ordering and equality relations are interpreted as usual. The logic $\mathrm{IA^{FO}}$ is a very expressive but highly undecidable framework for integer arithmetic since it contains predicates and functions with arbitrary arities together with the addition and multiplications operators.

Given an interpretation $M$, we evaluate a closed term $e$ into an integer $e^M$ to which the term simplifies when substituting every function symbol $f$ with their respective value $f^M(e_1^M, \cdots, e_m^M)$. The Boolean value $\phi^M$ of a closed formula $\phi$ can be determined in a similar way. An interpretation $M$ *satisfies* a formula $\phi$, denoted by $M \models \phi$, if $\phi^M = \top$. A set $\Phi$ of formulas is *satisfiable*, if there is an interpretation satisfying all formulas $\phi \in \Phi$. A set $\Phi$ of formulas *entails* a formula $\psi$, denoted by $\Phi \models \phi'$, if for all interpretations $M$ s.t. $M \models \phi$ for all formulas $\phi \in \Phi$, we have $M \models \psi$.

The validity problems of $\mathrm{IA^{FO}}$ is undecidable. We hereafter introduce a decidable fragment $\mathrm{LIA^P}$ of $\mathrm{IA^{FO}}$, where two restrictions are imposed: (1) it does not involve the multiplication operator and quantifier; and (2) all function and predicate symbols are 0-ary, which we call numeric variables.

Let $\mathcal{V}$ a finite set of numeric variables. The sets of *numeric terms* ($\mathtt{Term^P}$), *atomic formulas* ($\mathtt{Atom^P}$) and *formulas* ($\mathtt{Form^P}$) for $\mathrm{LIA^P}$ is defined by the following grammar:

$$e \in \mathtt{Term^P} :: c \mid F \mid e + e$$

$$\alpha \in \mathtt{Atom^P} :: e = e \mid e < e$$

$$\phi \in \mathtt{Form^P} :: \alpha \mid \neg\phi \mid \phi \wedge \phi$$

where $c \in \mathcal{Z}$ and $v \in \mathcal{V}$.

For the logic $\mathrm{LIA^P}$, its interpretation can be simplified as a valuation function, namely *state*, which maps each $v \in \mathcal{V}$ into $\mathcal{Z}$.

Throughout this paper, we fix an alphabet $\Delta$. A *string* is a finite sequence of characters over $\Delta$. We use $|\pi|$ for the length of $\pi$ and $\pi_i$ for the $i$-th symbol of $\pi$. A *substring* of $\pi$ is $\pi_i \circ \pi_{i+1} \circ \cdots \circ \pi_j$, denoted by $\pi_i^j$, where $1 \leq i \leq j \leq |\pi|$. The substring $\pi_1^i$ is a *prefix* of $\pi$ while $\pi_i^{|\pi|}$ is a *suffix*. A *subsequence* of $\pi$ is $\pi_{i_1} \circ \pi_{i_2} \circ \cdots \circ \pi_{i_j}$ where $1 \leq i_1 < i_2 < \cdots < i_j \leq |\pi|$. We remark that a substring is a subsequence, but not vice versa. For example, $aba$ is a subsequence of $abbba$, but it is not a substring.

The set of *regexes* ($\mathtt{Reg}$) is recursively defined by

$$r \in \mathtt{Reg} :: \varepsilon \mid a \mid r \circ r \mid (r \mid r) \mid r^*$$

where $\varepsilon$ denotes the empty string and $a \in \Delta$.

The regex $r_1 \circ r_2$ is called a *concatenation regex*, $r \mid r$ is called an *alternation regex*, and $r^*$ is called a *star regex*. We say $r$ is the *generator* of a star regex $r^*$.

The set of strings $\mathcal{L}(r)$ represented by $r$, is recursively defined as

- $\mathcal{L}(\varepsilon) = \{\varepsilon\}$ and $\mathcal{L}(a) = \{a\}$;
- $\mathcal{L}(r_1 \circ r_2) = \{s_1 s_2 \mid s_1 \in \mathcal{L}(r_1) \text{ and } s_2 \in \mathcal{L}(r_2)\}$;
- $\mathcal{L}(r_1 \mid r_2) = \mathcal{L}(r_1) \cup \mathcal{L}(r_2)$;
- $\mathcal{L}(r_1^*) = \mathcal{L}(\varepsilon) \cup \bigcup_{i \geq 1} \mathcal{L}(r_1^i)$ where $r_1^i = \overbrace{r_1 \cdot r_1 \cdots r_1}^{i}$.

A regex $r$ *accepts* a string $\pi$, if $\pi \in \mathcal{L}(r)$.

## Generalized Linear Integer Numeric Planning

In this section, we introduce concepts on linear integer numeric planning (LINP) based on $\mathrm{LIA^P}$, and then provide the definition of generalized LINP (GLINP) problems, and finally give a program-like definitions of solutions to GLINP problems.

**Definition 1.** A LINP domain $\mathcal{D}$ is a tuple $\langle \mathcal{V}, \mathcal{A} \rangle$ where

- $\mathcal{V}$: a finite set of numeric variables;
- $\mathcal{A}$: a finite set of actions which is defined by a tuple $\langle \mathtt{pre}, \mathtt{eff} \rangle$ where $\mathtt{pre} \in \mathtt{Form^P}$ is the precondition and $\mathtt{eff}$ is a finite set of numeric effects.

A *numeric effect* is a tuple $\langle v, e \rangle$ where $v \in \mathcal{V}$ and $e \in \mathtt{Term^P}$. Intuitively, it means that the numeric value of $v$ becomes $e^s$ after performing the action; otherwise, it remains unchanged. An action $a$ is *executable* in a state $s$, if $s \models \mathtt{pre}(a)$. The *successor state* of applying an action $a$ over $s$ is written as $\tau(s, a)$, which results from $s$ by mapping $v$ to $e^s$ (i.e. $\tau(s, a)(v) = e^s$) for all $\langle v, e \rangle \in \mathtt{eff}(a)$. We remark that $\tau(s, a)$ is still well-defined even if $a$ is non-executable in $s$.

The *resulting state* of performing a finite sequence $[a_1, \cdots, a_n]$ of actions on $s$ is recursively defined by $\tau(s, [a_1, \cdots, a_n]) = \tau(\tau(s, [a_1, \cdots, a_{n-1}]), a_n)$ and $\tau(s, \varepsilon) = s$ where $\varepsilon$ is an empty sequence. A (possibly infinite) sequence $[a_1, a_2, \cdots]$ of actions is *executable* in a state $s$, if $s \models \mathtt{pre}(a_1)$ and $\tau(s, [a_1 \cdots a_i]) \models \mathtt{pre}(a_{i+1})$ for $i \geq 1$.

A LINP problem is defined as a tuple $\langle \mathcal{D}, s_{\mathcal{I}}, \mathcal{G} \rangle$ where $\mathcal{D}$ is a LINP domain, $s_{\mathcal{I}}$ is an initial state, and $\mathcal{G} \in \text{Form}^{\text{P}}$ denoting a set of goal states. A solution to a LINP problem with an initial state $s$, namely *sequential plan*, is a finite sequence $[a_1, \cdots, a_n]$ of actions s.t. performing these actions one by one from $s$ leads to a final state satisfying the goal condition $\mathcal{G}$. More formally, $[a_1, \cdots, a_n]$ is executable in $s$, and $\tau(s, [a_1, \cdots, a_n]) \models \mathcal{G}$. We say $(s, \pi)$ a *state-plan pair* where $\pi$ is a sequential plan for $s$. Given a set $\Upsilon$ of state-plan pairs, we use $S(\Upsilon)$ for the set of states and $\Pi(\Upsilon)$ for the set of plans.

Generalized LINP (GLINP) problems are an extension to LINP problems that involve possibly infinite initial states that represented by a $\text{LIA}^{\text{P}}$ formula $\mathcal{I}$.

**Definition 2.** A generalized LINP (GLINP) problem $\Sigma$ is a tuple $\langle \mathcal{D}, \mathcal{I}, \mathcal{G} \rangle$, where

- $\mathcal{D}$: a LINP domain $\langle \mathcal{V}, \mathcal{A} \rangle$;
- $\mathcal{I} \in \text{Form}^{\text{P}}$: a formula representing a set of initial states.
- $\mathcal{G} \in \text{Form}^{\text{P}}$: a formula representing a set of goal states.

Each LINP problem $\langle \mathcal{D}, s_{\mathcal{I}}, \mathcal{G} \rangle$ is an instance of an GLINP problem $\langle \mathcal{D}, \mathcal{I}, \mathcal{G} \rangle$ for $s_{\mathcal{I}} \models \mathcal{I}$. It is easily verified that the existence of solutions to GLINP is undecidable from the undecidability result for LINP problems (Helmert 2002).

The solutions to GLINP problems is captured by *planning programs*, consisting of an empty plan, primitive actions, sequential, branch and loop structures.

**Definition 3** (Planning programs). The set of planning programs (Prog) for a linear integer numeric planning domain $\mathcal{D} = \langle \mathcal{V}, \mathcal{A} \rangle$ is recursively defined by

$\delta \in \text{Prog} :: \varepsilon \mid a \mid \delta; \delta \mid \textbf{if } \phi \textbf{ then } \delta \textbf{ else } \delta \textbf{ fi} \mid \textbf{while } \phi \textbf{ do } \delta \textbf{ od}$

where $a \in \mathcal{A}$ and $\phi \in \text{Form}^{\text{P}}$.

We say $\phi$ is the *condition* of the branch structure **if** $\phi$ **then** $\delta_1$ **else** $\delta_2$ **fi**. Likewise, $\phi$ is the condition of the loop structure **while** $\phi$ **do** $\delta$ **od**.

Given a GLINP problem, we are interested in synthesizing a program $\delta$ satisfying the following three critical properties of planning programs: (1) *termination*: $\delta$ will terminate eventually; (2) *reachability*: performing $\delta$ in an initial state leads to a goal state; (3) *executability*: any action is always executable during the execution of $\delta$.

**Definition 4.** Let $\mathcal{D} = \langle \mathcal{V}, \mathcal{A} \rangle$ be a LINP domain, $\delta$ a program for $\mathcal{D}$, and $s$ a state. The action sequence of executing $\delta$ in a state $s$ is defined as follows:

- $\Theta(s, \varepsilon) = \varepsilon$;
- $\Theta(s, a) = [a]$ where $a \in \mathcal{A}$.
- $\Theta(s, \delta_1; \delta_2) =$
  $$\begin{cases} \Theta(s, \delta_1) \circ \Theta(\tau(s, \delta_1), \delta_2), & \text{if } \Theta(s, \delta_1) \text{ is finite;} \\ \Theta(s, \delta_1), & \text{otherwise.} \end{cases}$$
- $\Theta(s, \textbf{if } \phi \textbf{ then } \delta_1 \textbf{ else } \delta_2 \textbf{ fi}) = \begin{cases} \Theta(s, \delta_1), & \text{if } \phi(s) = \top; \\ \Theta(s, \delta_2), & \text{otherwise.} \end{cases}$
- $\Theta(s, \textbf{while } \phi \textbf{ do } \delta_1 \textbf{ od}) =$
  $$\begin{cases} \Theta(s, \delta_1) \circ \Theta(\tau(s, \delta_1), \textbf{while } \phi \textbf{ do } \delta_1 \textbf{ od}), \\ \qquad \text{if } \phi(s) = \top \text{ and } \Theta(s, \delta_1) \text{ is finite;} \\ \Theta(s, \delta_1), \quad \text{if } \phi(s) = \top \text{ and } \Theta(s, \delta_1) \text{ is infinite;} \\ \varepsilon, \qquad\qquad \text{otherwise.} \end{cases}$$

---

**Algorithm 1:** SynPlanProg($\Sigma$)

**Input:** $\Sigma$: A GLINP Problem
**Output:** $\delta$: a planning program that is a solution to $\Sigma$
**1** Initialize a bound $b$
**2** $S \leftarrow$ GenInitStates($\Sigma, b$)
**3** $(r, \Upsilon) \leftarrow$ GenSkeleton($\Sigma, S$)
**4** $\delta \leftarrow$ Complete($r, \Upsilon$)
**5 return** $\delta$

---

where $\tau(s, \delta)$ is $\tau(s, \Theta(s, \delta))$ if $\Theta(s, \delta)$ is finite and $\Theta_1 \circ \Theta_2$ is the concatenation of two sequences of actions $\Theta_1$ and $\Theta_2$.

**Definition 5.** Let $\mathcal{D} = \langle \mathcal{V}, \mathcal{A} \rangle$ be a LINP domain, $\delta$ a program for $\mathcal{D}$ and $s$ a state. The program $\delta$ is

- *terminable* in $s$, iff $\Theta(s, \delta)$ is finite;
- *executable* in $s$, iff $\Theta(s, \delta)$ is executable in $s$;
- *$\phi$-reachable* in $s$, iff $\delta$ is terminable and executable in $s$ only if $\tau(s, \delta) \models \phi$.

A planning program is a solution to a GLINP problem, if it satisfies the above three properties for any initial state.

**Definition 6.** Let $\Sigma = \langle \mathcal{D}, \mathcal{I}, \mathcal{G} \rangle$ be a GLINP problem and $\delta$ a planning program for $\mathcal{D}$. The program $\delta$ is a solution to $\Sigma$, if for any state $s \models \mathcal{I}$, we have $\delta$ is $\mathcal{G}$-reachable, terminable and executable in $s$.

## The Main Algorithm

We now present a generation approach to synthesize planning programs for a given domain $\Sigma$ shown in Algorithm 1. The main idea of Algorithm 1 is to construct a candidate planning program $\delta$ which can satisfy with all instances of the initial states set

The notion of regexes is highly related to planning programs. Suppose that the alphabet $\Delta$ is the set of actions $\mathcal{A}$ of a given domain $\Sigma$. A sequence $\pi$ of actions is a string over $\Delta$. Each construct of a regex corresponds to a structure of planning programs. If the condition of the branch structure **if** $\phi$ **then** $\delta_1$ **else** $\delta_2$ **fi** is omitted, then it corresponds to a concatenation of $\delta_1$ and $\delta_2$. Similarly, the loop structure **while** $\phi$ **do** $\delta$ **od** corresponds to the star regex $\delta^*$ if the condition is left out. Hence, regexes can be considered as skeletons of planning programs.

Inspired by the intimate connection between regexes and planning programs, we divide the generation stage into two steps: (1) synthesize a skeleton of planning program $r$ expressed in a regex, and generate a set of state-plan pairs $\Upsilon$ where each pair $(s, \pi)$ denotes $\pi$ is a sequential plan for $s$ (Line 3); (2) obtain a complete planning program $\delta$ by filling the missing conditions in $r$ according to $\Upsilon$ (Line 5). Then the test stage will check the validity of the planning program $\delta$ (Line 6). If $\delta$ is correct, then it returns $\delta$ as the solution (Lines 7 and 8). Otherwise, it returns an initial state $s$ such that executing $\delta$ in $s$ cannot reach the goal. In this case, a sequential plan $\pi$ corresponding to $s$ will be computed by a numeric planner (Line 10). If $r$ does not accept the sequence of actions $\pi$, then we consider the skeleton $r$ is incorrect, and restart the generation stage with a larger bound $b$ (Lines

**Algorithm 2:** GenSkeleton($\Sigma, S$)

---

**Input:** $\Sigma$: the planning problem $\Sigma$
$\quad\quad$ $S$: the finite set of states
**Output:** $r$: a skeleton of planning programs
$\quad\quad\quad$ $\Upsilon$: the set of state-plan pairs

**1** $\Upsilon \leftarrow \emptyset$ **foreach** $s \in S$ **do**
**2** $\quad$ $\pi \leftarrow$ Plan($\Sigma, s$)
**3** $\quad$ $\Upsilon \leftarrow \Upsilon \cup \{(s, \pi)\}$
**4** $R \leftarrow \emptyset$ and $\Delta^* \leftarrow \emptyset$
**5** **foreach** $\pi \in \Pi(\Upsilon)$ **do**
**6** $\quad$ $t \leftarrow \pi$ and $\Delta \leftarrow \mathcal{A}$
**7** $\quad$ **while** $true$ **do**
**8** $\quad\quad$ $(t', \Delta') \leftarrow$ FoldString($t, \Delta, 1$)
**9** $\quad\quad$ **if** $t \neq t'$ **then**
**10** $\quad\quad\quad$ $\Delta \leftarrow \Delta \cup \Delta'$ and $t \leftarrow t'$
**11** $\quad\quad$ **else**
**12** $\quad\quad\quad$ break;
**13** $\quad$ $R \leftarrow R \cup \{t\}$ and $\Delta^* \leftarrow \Delta^* \cup \Delta$
**14** $(\xi_1, \cdots, \xi_l) \leftarrow$ the sequence of non-extensible common strings of $R$ over $\Delta^*$
**15** Compute each $i$-th individual components $\eta_{i,j}$ of $t_j$ s.t.
$\quad\quad$ $t_j = \eta_{1,j} \circ \xi_1 \circ \cdots \circ \eta_{l,j} \circ \xi_l \circ \eta_{l+1,j}$ for $1 \leq i \leq l+1$
$\quad\quad$ and $1 \leq j \leq k$
**16** $\eta_i \leftarrow \eta_{i,1} \mid \cdots \mid \eta_{i,k}$ for $1 \leq i \leq l+1$
**17** $r \leftarrow \eta_1 \circ \xi_1 \circ \eta_2 \circ \xi_2 \circ \cdots \circ \eta_l \circ \xi_l \circ \eta_{l+1}$
**18** Simplify $r$

---

11 - 13). Otherwise, some conditions in $r$ is incorrect, and hence the set of state-plan pairs $\Upsilon$ is enlarged by the pair $(s, \pi)$, and complete the conditions occurring in $r$ again.

The main algorithm consists of 3 procedures: GenInitStates, GenSkeleton, Complete. The three procedures will be sequentially explained in the following sections.

## Generation of Initial States and Skeletons of Planning Programs

Given a planning domain $\mathcal{D}$, the GenSkeleton procedure (Algorithm 2) aims to guess a suitable skeleton of planning programs for $\mathcal{D}$ expressed by a regex $r$. The main insight behind the procedure is to infer a regex based on a set of strings (*i.e.* a set of sequential plans).

In the area of grammatical inference, Kinber (2010) proposed a learning algorithm to infer a regex $r$ with star operators from one string. To facilitate identifying the star subregex, Kinber (2010) requires the given string $s$ to be representative for the regex $r$, more formally, the generator of any star subregex of $r$ consecutively occurs in the string $s$ at least twice. We observe from most planning domains that if the values of all numeric variables of an initial state $s$ are large enough, then the corresponding plan is representative. To assist the GenSkeleton procedure, the GenInitState procedure initializes a set $S$ of initial states where the absolute values of all numeric variables are at least as large as the bound $b$.

With the set of initial states in hand, the GenSkeleton procedure works as follows. It firstly invokes the Plan procedure to compute the sequential plan for every state $s \in S$ (Lines 1 - 4).

The second step of the GenSkeleton procedure is to fold each representative plan $\pi \in \Pi$ into a regex $t$ with star operators (Lines 6 - 16). It starts from the original alphabet $\Delta = \mathcal{A}$ (Line 8), and infers a regex $t'$ over $\Delta$ accepting $\pi$ (Line 10). If the current regex $t'$ is not equal to the previous one $t$, it means that new star subregexes are found in this iteration. These subregexes are considered as new single characters that are used to enlarge the alphabet $\Delta$. Hence $t'$ can be considered as a string over the new alphabet $\Delta$. The above computations will continue until no new star subregexes are identified (*i.e.* the regex remains unchanged, $t = t'$). The

---

**Algorithm 3:** FoldString($\Delta, \pi, l$)

---

**Input:** $\Delta$: the alphabet
$\quad\quad$ $\pi$: a string
$\quad\quad$ $l$: the length of the generator of a star subregex
**Output:** $r$: a star regex
$\quad\quad\quad$ $\Delta'$: the extended alphabet that contains
$\quad\quad\quad\quad$ the identified star subregex

**1** $r \leftarrow \varepsilon$ and $\Delta' \leftarrow \emptyset$
**2** $i, j \leftarrow 1$
**3** **while** $j < |\pi|$ **do**
**4** $\quad$ **if** $\pi[j] \cdots \pi[j+l] = \pi[j+l+1] \cdots \pi[j+2l]$ **then**
**5** $\quad\quad$ $\pi' \leftarrow \pi[j] \cdots \pi[j+l]$
**6** $\quad\quad$ **if** $j - i \geq 2l+2$ **then**
**7** $\quad\quad\quad$ $(r', \Delta'_{l+1}) \leftarrow$
$\quad\quad\quad\quad$ FoldString($\Delta, \pi[i] \cdots \pi[j], l+1$)
**8** $\quad\quad\quad$ $\Delta' \leftarrow \Delta' \cup \Delta'_{l+1} \cup \{(\pi')^*\}$
**9** $\quad\quad\quad$ $r \leftarrow r \cdot r' \cdot (\pi')^*$
**10** $\quad\quad$ **else**
**11** $\quad\quad\quad$ $r \leftarrow r \cdot (\pi[i] \cdots \pi[j]) \cdot (\pi')^*$
**12** $\quad\quad\quad$ $\Delta' \leftarrow \Delta' \cup \{(\pi')^*\}$
**13** $\quad\quad$ Let $k$ be the time of occurrences of $\pi'$ in the longest substring $(\pi')^k$ of $\pi$ beginning at $j$ (??)
**14** $\quad\quad$ $i, j \leftarrow j + kl + 1$
**15** $\quad\quad$ **if** $\exists j' > j, \pi[j] \cdots \pi[j']$ *is accepted by* $\pi'$ *and*
$\quad\quad\quad$ $\forall i' > j', \pi[j] \cdots \pi[i']$ *is not accepted by* $\pi'$ **then**
**16** $\quad\quad\quad$ $i, j \leftarrow j'$
**17** $\quad$ **else**
**18** $\quad\quad$ $j \leftarrow j + 1$
**19** **if** $|\pi| - i \geq 2l+2$ **then**
**20** $\quad$ $(r', \Delta'_{l+1}) \leftarrow$ FoldString($\pi[i] \cdots \pi[j], \Delta, l+1$)
**21** $\quad$ $\Delta' \leftarrow \Delta' \cup \Delta'_{l+1}$
**22** $\quad$ $r \leftarrow r \cdot r'$
**23** **else**
**24** $\quad$ $r \leftarrow r \cdot (\pi[i] \cdots \pi[j])$
**25** **return** $r, \Delta'$

---

FoldString process (Algorithm 3) generates a regex with star subregexes $u^*$ according to $t$. It starts recognizing $u^*$ where the length of the generator $u$ is 1, and then continues to handle larger cases by increasing the length of generators. When the length of generator is more than the half of the length of $t$, the recognization process terminates since it is impossible to find the new star subregexes. The regex $t$ is gathered into the set $R$ and the alphabet $\Delta^*$ is enlarged by $\Delta$ (Line 15). At each iteration, it identifies the star subregex $u^*$ when the substring $u$ over $\Delta$ consecutively occurs in $t$ at least twice, and then replaces all of the longest substrings $u \cdots u$ in $t$ by $u^*$.

Finally, the GenSkeleton procedure merges all regexes of $R$ into a final regex $r$ with the alternation connective based on the notion of common substrings and common subsequences (Lines 16 - 20). A string $\xi$ is a *common substring* of $R$, if $\xi$ is a substring of $t_j \in R$ for $1 \leq j \leq k$. Similarly, a string $\xi$ is a *common subsequence* of $R$, if $\xi$ is a subsequence of $t_j \in R$ for $1 \leq j \leq k$. The sequence of non-extensible common strings of $R$ is $(\xi_1, \cdots \xi_l)$ s.t. $\xi_1 \circ \cdots \circ \xi_l$ is the longest common subsequence of $R$, and $\xi_i$ is a non-extensible common substring of $\Pi$ (more precisely, $(\xi_{i-1})_{|\xi|} \circ \xi_i$ is not a common substring of $R$ for $i > 1$, and $\xi_i \circ (\xi_{i+1})_1$ is a not common substring of $\Pi$ for $i < l$). Each $t_j \in R$ is the concatenation $\eta_{1,j} \circ \xi_1 \circ \cdots \circ \eta_{l,j} \circ \xi_l \circ \eta_{l+1,j}$ where $\eta_{1,j}$ and $\eta_{l+1,j}$ may be the empty string $\varepsilon$. Each $\eta_{i,j}$ is the $i$-th individual component of $t_j$ while $\xi_i$ is the common component of the $R$ for $1 \leq i \leq l+1$ and $1 \leq j \leq k$. We remind that each regex $t$ of $R$ is a string over $\Delta^*$. The final regex $r$ is obtained as follows. We firstly generate the sequence of common strings of $R$ is $(\xi_1, \cdots \xi_l)$ and each $i$-th individual component of $t_j$ (Lines 16 and 17). We then obtain the regex $\eta_i$ via merging all of the $i$-th independent component of $t_j$'s via alternation connectives (*i.e.* $\eta_{i,1} \mid \cdots \mid \eta_{i,k}$) for each $1 \leq i \leq l+1$ (Line 18). We concatenate the combination of individual components $\eta_i$ and the common component $\xi_i$ alternatively in an increasing order (Line 19). In the end, some redundant subregexes in $r$ are removed (Line 20). For example, $u_1 \mid u_2 \mid \cdots \mid u_{k-1} \mid u_2 \mid u_k$ is simplified as $u_1 \mid u_2 \mid \cdots \mid u_{k-1} \mid u_k$.

**Theorem 1.** *Let $\Sigma = \langle \mathcal{D}, \mathcal{I}, \mathcal{G} \rangle$ be a GLINP problem and $S$ a finite set of initial state. Let $(r, \Upsilon)$ be the output of* GenSkeleton$(\Sigma, S)$ *where $r$ is the skeleton of planning programs and $\Upsilon$ is the set of state-plan pairs. Assume that the procedure* Plan *always returns a plan $\pi$ for every initial state $s \in S$. Then, $S = S(\Upsilon)$, and $r$ is the regex accepting each $\pi \in \Pi(\Upsilon)$.*

*Proof sketch:* The first half of the theorem can be directly obtained from the assumption that the procedure Plan always computes a plan $\pi$ for every initial state $s \in S$, and each state-plan pair $(s, \pi)$ is added into the set $\Upsilon$.

Now we prove the second half of the theorem. For each $\pi_j \in \Pi(\Upsilon)$, the FoldString process iteratively replaces the consecutive occurrence of substring $u$ in $\pi_j$ by $u^*$, and obtain the regex $t_j$. So each $t_j$, which is added into the set $R$, accepts $\pi_j$. Then, each $t_j$ is divided into a sequence of subregexes (*i.e.* , $\eta_{1,j} \circ \xi_1 \circ \cdots \circ \eta_{l,j} \circ \xi_l \circ \eta_{l+1,j}$ where $\eta_{i,j}$ is an individual component of $t_j$ and $\xi_i$ is a common component of the $R$). The final regex $r$ is $\eta_1 \circ \xi_1 \circ \eta_2 \circ \xi_2 \circ \cdots \circ \eta_l \circ \xi_l \circ \eta_{l+1}$ where $\eta_i = \eta_{i,1} \mid \cdots \mid \eta_{i,k}$. Obviously, $r$ accepts every $\pi_i \in \Pi(\Upsilon)$. $\qquad \square$

## Completion of Planning Programs

Now we are ready to construct a candidate planning program by completing the branch and loop conditions occurring in the skeleton $r$ generated in the above section.

We adapt the enumerative algorithm proposed by Udupa et al. (2013), to infer these conditions. Given a set $S^+$ of positive states and a set $S^-$ of negative states, the main idea of the enumerative algorithm is to iteratively generates the

---

**Algorithm 4:** Complete$(r, \Upsilon)$

**Input:** $r$: a skeleton of planning program
$\Upsilon$: a set of state-plan pairs
**Output:** $\delta$: a complete planning program

1 **switch** $r$ **do**
2    **case** $r_1 \mid r_2$ **do**
3      $S^+ \leftarrow \emptyset$ and $S^- \leftarrow \emptyset$
4      $\Upsilon_1 \leftarrow \emptyset$ and $\Upsilon_2 \leftarrow \emptyset$
5      **foreach** $(s, \pi) \in \Upsilon$ **do**
6        **if** $\pi \in \mathcal{L}(r_1)$ **then**
7          $S^+ \leftarrow S^+ \cup \{s\}$
8          $\Upsilon_1 \leftarrow \Upsilon_1 \cup \{(s, \pi)\}$
9        **else**    /* $\pi \in \mathcal{L}(r_2)$ */
10          $S^- \leftarrow S^- \cup \{s\}$
11          $\Upsilon_2 \leftarrow \Upsilon_2 \cup \{(s, \pi)\}$
12      $\delta_1 \leftarrow$ Complete$(r_1, \Upsilon_1)$
13      $\delta_2 \leftarrow$ Complete$(r_2, \Upsilon_2)$
14      $\phi \leftarrow$ Enumerate$(S^+, S^-)$
15      $\delta \leftarrow$ **if** $\phi$ **then** $\delta_1$ **else** $\delta_2$ **fi**
16    **case** $r_1^*$ **do**
17      $S^+ \leftarrow \emptyset$ and $S^- \leftarrow \emptyset$
18      $\Upsilon_1 \leftarrow \emptyset$
19      **foreach** $(s, \pi) \in \Upsilon$ **do**
20        Computes substrings $(\pi_1, \pi_2, \cdots, \pi_k)$ s.t. $\pi_1 \circ \pi_2 \circ \cdots \circ \pi_k = \pi$ and $\pi_i \in \mathcal{L}(r_1)$ for $1 \leq i \leq k$;
21        **for** $i \leftarrow 1$ **to** $k$ **do**
22          $S^+ \leftarrow S^+ \cup \{s\}$
23          $\Upsilon_1 \leftarrow \Upsilon_1 \cup \{(s, \pi_i)\}$
24          $s \leftarrow \tau(s, \pi_i)$
25        $S^- \leftarrow S^- \cup \{s\}$
26      $\delta_1 \leftarrow$ Complete$(r_1, \Upsilon_1)$
27      $\phi \leftarrow$ Enumerate$(S^+, S^-)$
28      $\delta \leftarrow$ **while** $\phi$ **do** $\delta_1$ **od**
29    **case** $r_1 \circ r_2$ **do**
30      $\Upsilon_1 \leftarrow \emptyset$ and $\Upsilon_2 \leftarrow \emptyset$
31      **foreach** $(s, \pi) \in \Upsilon$ **do**
32        Computes the prefix $\pi_1^i$ and suffix $\pi_{i+1}^{|\pi|}$ of $\pi$ s.t. $\pi_1^i \in \mathcal{L}(r_1)$ and $\pi_{i+1}^{|\pi|} \in \mathcal{L}(r_2)$
33        $\Upsilon_1 \leftarrow \Upsilon_1 \cup \{(s, \pi_1^i)\}$
34        $\Upsilon_2 \leftarrow \Upsilon_2 \cup \{(\tau(s, \pi_1^i), \pi_{i+1}^{|\pi|})\}$
35      $\delta_1 \leftarrow$ Complete$(r_1, \Upsilon_1)$
36      $\delta_2 \leftarrow$ Complete$(r_2, \Upsilon_2)$
37      $\delta \leftarrow \delta_1; \delta_2$
38    **otherwise do**    /* $r = \varepsilon$ or $r = a$ */
39      $\delta \leftarrow r$

---

candidate formulas by induction on size until it finds an excepted formula $\phi$ that is consistent with the two sets $S^+$ and $S^-$, more formally, $s \models \phi$ for $s \in S^+$ and $s \models \neg\phi$ for $s \in S^-$. For details, please refer to Algorithm 1 proposed in this paper (Udupa et al. 2013).

It remains to construct the set of positive states and the set of negative states for each condition of branch structures and loop structures.

In the following, we illustrate the Complete procedure which takes a skeleton $r$, a set of state-plan pairs $\Upsilon$ as input, and outputs a complete planning program $\delta$. The

Complete procedure works in a recursive way. Suppose that the original regex is $r$.

In the case where $r = r_1 \mid r_2$ (Lines 2 - 15). This regex corresponds to the branch structure **if** $\phi$ **then** $\delta_1$ **else** $\delta_2$ **fi** where $\phi$ is the condition and the subregexes $r_1$ and $r_2$ correspond to $\delta_1$ and $\delta_2$, respectively. The set $S^+$ is the set of positive states for synthesizing $\phi$ and $S^-$ is that of negative states. The sets $\Upsilon_1$ and $\Upsilon_2$ are two sets of state-plan pairs for completing $r_1$ and $r_2$, respectively Initialy, the above four sets are empty (Lines 3 and 4). For each $(s, \pi) \in \Upsilon$, if $r_1$ accepts $\pi$, then the execution of $\pi$ enters the branch expressed by $r_1$, and hence the state $s$ is the positive state of $\phi$ (*i.e.* $s \models \phi$). The set $S^+$ is enlarged by $s$, and the state-plan pair $(s, \pi)$ is added into $\Upsilon_2$ (Lines 6 - 8). Otherwise, the execution of $\pi$ goes to another branch described by $r_2$. The state $s$ is the negative state of $\phi$ (*i.e.* $s \models \neg\phi$). Similarly to the opposite case, the set $S^-$ is enlarged by $s$, and $(s, \pi)$ is added into $\Upsilon_2$ (Lines 9 - 11). Then, we obtain the subprograms $\delta_1$ and $\delta_2$ by invoking Complete$(r_1, \Upsilon_1)$ and Complete$(r_2, \Upsilon_2)$, respectively, and construct the condition $\phi$ via the Enumerate process (Lines 12 and 13). Finally, the Complete procedure constructs the condition $\phi$ via the Enumerate process based on the set of positive states $S^+$ and the set of negative states $S^-$ (Line 14).

In the case where $r = r_1^*$ (Lines 17 - 28). This regex corresponds to the loop structure **while** $\phi$ **do** $\delta_1$ **od** where $\phi$ is the condition and the subregex $r_1$ corresponds to the program $\delta_1$. The meaning of $S^+$, $S^-$ and $\Upsilon_1$ are similar to the above case, and they are initialized as empty (Lines 17 and 18). For each $(s, \pi) \in \Upsilon$, if the sequential plan $\pi$ is not an empty plan, then there is a partition $(\pi_1, \pi_2, \cdots, \pi_k)$ s.t. their concatenation is $\pi$ and $r_1$ accepts each $\pi_i$ for $1 \leq i \leq k$ (Line 20). In other words, the program expressed by $r_1$ is executed by $k$ times, where the action sequence of the $i$-th execution is $\pi_i$. It is easily observed that the execution of the loop structure enters the body $\delta_1$ in the following states: $s, \tau(s, \pi_1), \cdots, \tau(s, \pi_1 \cdots \pi_{k-1})$. So the above states are the positve state of the condition $\phi$ and are added into $S^+$ (Line 22). Meanwhile, the following state-plan pairs $(s, \pi_1), (\tau(s, \pi_1), \pi_2), \cdots, (\tau(s, \pi_1 \cdots \pi_{k-1}), \pi_k)$ is added into $\Upsilon_1$ (Line 23). When the loop is completed, the execution is out of the loop and terminates in the state $\tau(s, \pi)$. We consider this state as the negative state of $\phi$ which is added into $S^-$ (Line 25). Then, the subprogram $\delta_1$ is obtained according to $r_1$ and $\Upsilon_1$ (Line 26). Finally, the Enumerate procedure synthesizes the condition $\phi$ according to $S^+$ and $S^-$ (Line 27).

In the case where $r = r_1 \circ r_2$ (Lines 29 - 37). This regex corresponds to the sequential structure $\delta_1; \delta_2$ where the subregexes $r_1$ and $r_2$ correspond to the program $\delta_1$ and $\delta_2$, respectively. The meaning of $\Upsilon_1$ and $\Upsilon_2$ are similar to the branch structure, and they are initialized as empty (Line 30). For each $(s, \pi) \in \Upsilon$, there are a prefix $\pi_1^i$ of $\pi$ and a suffix $\pi_{i+1}^{|\pi|}$ s.t. $r_1$ accepts the former and $r_2$ accepts the latter (Line 32). We collect the state-plan pair $(s, \pi_1^i)$ and $(\tau(s, \pi_1^i), \pi_{i+1}^{|\pi|})$ for $\Upsilon_1$ and $\Upsilon_2$, respectively (Lines 33 and 34). Finally, the subprograms $\delta_1$ and $\delta_2$ are constructed recursively (Lines 35 and 36).

The last case where $r = \varepsilon$ or $r = a$ can be easily handled. We obtain the soundness for the Complete procedure.

**Theorem 2.** *Let* $\Sigma = \langle \mathcal{D}, \mathcal{I}, \mathcal{G} \rangle$. *Let* $r$ *be a skeleton of a planning program of* $\mathcal{D}$ *and* $\Upsilon$ *a set of state-plan pairs where every state* $s \in S(\Upsilon)$ *is an initial state of* $\Sigma$ *and* $r$ *accepts each plan* $\pi$ *where* $\pi \in \Pi(\Upsilon)$. *Assume that the procedure* Plan *always returns a plan* $\pi$ *for every initial state* $s \in S$. *Let* $\delta$ *be the output of* Complete$(r, \Upsilon)$. *Then,* $\delta$ *is terminable, executable and* $\mathcal{G}$-*reachable in all states* $s \in S(\Upsilon)$.

*Proof.* We firstly prove that $\pi = \Theta(s, \delta)$ for $(s, \pi) \in \Upsilon$. We prove by induction on $\delta$.

- $\delta = \varepsilon$ or $\delta = a$: Suppose that $\delta = \varepsilon$. Since $r$ accepts $\pi$, $\pi = \varepsilon$, and hence $\Theta(s, \delta) = \varepsilon = \pi$. Similarly, $\Theta(s, a) = [a] = \pi$ when $\delta = a$.

- $\delta = \delta_1; \delta_2$: It follows that $r = r_1 \circ r_2$ where $r_i$ corresponds to the program $\delta_i$ for $i = 1, 2$. Since $r$ accepts $\pi$, there is an index $i$ s.t. $r_1$ accepts $\pi_1^i$ and $r_2$ accepts $\pi_{i+1}^{|\pi|}$. According to the Complete procedure (Algorithm 4), $(s, \pi_1^i) \in \Upsilon_1$ and $(\tau(s, \pi_1^i), \pi_{i+1}^{|\pi|}) \in \Upsilon_2$. By the inductive assumption, the Complete procedure synthesizes two programs $\delta_1$ and $\delta_2$ s.t. $\pi_1^i = \Theta(s, \delta_1)$ and $\pi_{i+1}^{|\pi|} = \Theta(\tau(s, \pi_1^i), \delta_2)$. Hence, $\pi = \pi_1^i \circ \pi_{i+1}^{|\pi|} = \Theta(s, \delta_1) \circ \Theta(\tau(s, \pi_1^i), \delta_2) = \Theta(s, \delta_1; \delta_2)$.

- $\delta = $ **if** $\phi$ **then** $\delta_1$ **else** $\delta_2$ **fi**: It follows that $r = r_1 \mid r_2$ where $r_i$ corresponds to the program $\delta_i$ for $i = 1, 2$. Suppose that $\phi^s = \top$. Thus, $\Theta(s, \delta) = \Theta(s, \delta_1)$. Since the Enumerate procedure constructs the condition $\phi$ consistent with $S^+$ and $S^-$ (*i.e.* $s^+ \models \phi$ for $s^+ \in S^+$ and $s^- \models \neg\phi$ for $s^- \in S^-$). So $s \in S^+$ and $(s, \pi) \in \Upsilon_1$. By the inductive assumption, the Complete procedure synthesizes a program $\delta_1$ s.t. $\pi = \Theta(s, \delta_1)$. Hence, $\pi = \Theta(s, \delta)$. The case where $\phi^s = \bot$ can be similarly proved.

- $\delta = $ **while** $\phi$ **do** $\delta_1$ **od**: It follows that $r = r_1^*$ where $r_1$ corresponds to the program $\delta_1$. Suppose that $\pi \neq \varepsilon$. The case where $\pi = \emptyset$ is easier. Since $r$ accepts $\pi$, there is a sequence of substrings $(\pi_1, \pi_2, \cdots, \pi_k)$ s.t. $\pi_1 \circ \pi_2 \circ \cdots \circ \pi_k = \pi$ and $\pi_i \in \mathcal{L}(r_1)$ for $1 \leq i \leq k$. According to the Complete procedure, $(s, \pi_1) \in \Upsilon_1$ and $(\tau(s, \pi_1 \circ \cdots \circ \pi_i), \pi_{i+1}) \in \Upsilon_1$ for $1 \leq i < k$. By the inductive assumption, the Complete procedure synthesizes a program $\delta_1$ s.t. $\pi_1 = \Theta(s, \delta_1)$ and $\pi_{i+1} = \Theta(\tau(s, \pi_1 \circ \cdots \circ \pi_i), \delta_1)$ for $1 \leq i < k$. In addition, $s \in S^+$, $\tau(s, \pi_1 \circ \pi_i) \in S^+$, and $\tau(s, \pi_1 \circ \pi_k) \in S^-$. So $s^+ \models \phi$ for $s^+ \in S^+$ and $s^- \models \neg\phi$ for $s^- \in S^-$. By Definition 4, we get that $\Theta(s, $ **while** $\phi$ **do** $\delta_1$ **od**$) = \Theta(s, \delta_1) \circ \cdots \circ \Theta(\tau(s, \pi_1 \circ \cdots \circ \pi_{k-1}), \delta_1)$. Hence, $\pi = \Theta(s, $ **while** $\phi$ **do** $\delta_1$ **od**$)$.

Since $\pi$ is also a solution to the LINP problem $\langle \mathcal{D}, s, \mathcal{G} \rangle$ for $(s, \pi) \in \Pi$, $\pi$ is finite and executable in $s$, and $\tau(s, \pi) \models \mathcal{G}$. This, together with the fact that $\pi = \Theta(s, \delta)$, imply $\delta$ is a terminable, executable and $\mathcal{G}$-reachable in $s$. $\square$

| Domain | Variables | | Examples | Loop | Length | GenSkeleton (s) | Completion (s) | Total (s) |
|--------|---|---|----------|------|--------|-----------------|----------------|-----------|
| | P | N | | | | | | |
| Chop | 0 | 1 | 3 | 1 | 5 | 0.037 | 0.002 | 0.039 |
| Gripper | 3 | 2 | 3 | 1 | 11 | 0.089 | 0.005 | 0.094 |
| Arith | 0 | 2 | 3 | 1 | 15 | 0.051 | 0.097 | 0.148 |
| Corner-A | 0 | 4 | 3 | 1 | 11 | 0.062 | 0.018 | 0.080 |
| D-Return | 4 | 6 | 3 | 1 | 40 | 0.187 | 0.084 | 0.281 |
| Delivery | 3 | 2 | 3 | 1 | 15 | 0.073 | 0.005 | 0.078 |
| Snow | 2 | 2 | 3 | 1 | 15 | 0.061 | 0.008 | 0.069 |
| Hall-A | 4 | 5 | 3 | 1 | 37 | 0.143 | 0.036 | 0.179 |
| NestVar | 0 | 2 | 3 | 2 | 11 | 0.049 | 0.009 | 0.058 |
| VisitAll | 2 | 5 | 3 | 2 | 21 | 0.105 | 0.036 | 0.141 |
| Sailing | 0 | 2 | 3 | 1 | 7 | 0.055 | 0.253 | 0.308 |

Table 1: Experimental results: the column "Variables" denotes the number of variables where "P" denotes the number of propositional variables and "N" denotes the number of numeric variables. The column "Examples" is the total number of examples, "Loop" denotes the depth of loop in the planning programs, "Length" denotes the length of the planning program. The column "GenSkeleton" denotes the runtime of generation of skeletons of planning programs, "Completion" denotes the runtime of completion of planning programs, "Total" denotes the Total runtime.

## Experimental Evaluation

We have performed a sets of experiments for synthesizing the generalized planning programs, corresponding to skelton generation, completion and verfication. Most domains[1] used in the experiments are the same as in other previous work, including: Arith (Hu and Levesque 2009), Chop (Levesque 2005), Corner-A, D-Return, Hall-A, Prize-A, Gripper (Bonet, Palacios, and Geffner 2009), Delivery (Srivastava, Immerman, and Zilberstein 2011), NestVar, Snow (Srivastava et al. 2011b), and Sailing (Scala, Haslum, and Thiébaux 2016). In Arith, the aim is to make the value of $v1$ (resp. $v2$) to an integer number $n$ (resp. $2n + 1$). In Chop, the aim is to decrease the height of tree. In Corner-A, the aim is to reach the top-right corner from any position in grid. In D-Return, the aim is to is to reach the bottom-right corner, the top-right corner, and bottom-right corner in order, and finally return to the starting position. In VistiAll, the aim is to visit all cells of grid starting at the top-left corner. In Hall-A, the aim is to visit the four corners of a grid. In Snow, the aim is to sort the elements of a vector. In Delivery, the aim is to transport all packages to a company with a truck of capacity one. Finally, in NestVar, the aim is to make the value of $x$ to 0, where two values $x$ and $y$ with positive integer numbers initially. We implemented the proposed approach in a system by using Python, Z3 (de Moura and Bjørner 2008) and Metric-FF (Hoffmann 2003). All experiments were conducted on a machine with an Intel Core i5 2.50 GHz CPU and 8GB RAM under Ubuntu 16.04. We found that the generated planning solution can satisfy all the (infinite) instance of the domain by manual verification. Namely, the generated planning solution is total correct, satisfying with the three critical properties: reachability, executability and termination.

In each domain, except for Chop, Arith, Cornner-A, NestVar, Sailing whose domain comprises just the numeric variables, the definition of other domains requires propositional variables. For these domains, we observe a

characteristic: their propositional variables actually have only true and false assignments in the effect of all their actions. So in essence, it can be regarded as a numerical variable. When the value of the numerical variable is 1, the corresponding is true. Otherwise, when it is false, it corresponds to 0. Therefore, when the propositional variable conforms to the above rules, the corresponding domain is also applicable to our approach. Furthermore, we observe that the above rules of propositional variables satisfies in many domains of generalized planning synthesis.

Finally, we close this section by summarizing experimental results. The summary of the experimental results is shown in Table 1, from which we can make some observations. Firstly, our approach is able to solve all domains in a reasonable amount of time ($<$ 1s). This justifies the effectiveness and scalability of our approach. Hence our approach provides an effective way of constructing solutions to GLINP. Secondly, most of the domains spend more than half of time in generating skeletons of planning programs except the sailing and Arith problems, with most of the time spent in completing planning programs. This is because the conditions of planning programs of these two problems are more complicated than other ones. Therefore, when infering to the corresponding conditions, it takes a longer time for enumeration synthesis. Thirdly, for D-Return and Hall-A problems, the generation procedure is quite time-consuming. And the main reason is that the length of the planning programs are relatively large, and hence the corresponding generating skeletons are significantly longer than other ones. Moreover, for the generation of the nested loop planning program, the time required is mainly spent on the synthesis of the nested skelton.

## Related Work

Qualitative numeric planning (QNP), proposed in (Srivastava et al. 2011b), is very close to our work. It is a class of numeric planning with many initial states with the assumption that all states are fully observable. Two restrictions are imposed on numeric planning: (1) the formula to be a form

---

[1]Some domains involves first-order predicates and they are modified so that it can be formalized in integer numeric planning.

of $v > 0$ or $v = 0$; and (2) the effects of actions decrease or increase the value of some variables by an unspecified amount. Under these two restrictions, the state space can be compressed into a finite space. More precisely, the space contains $2^{|\mathcal{V}|}$ states where $|\mathcal{V}|$ is the number of numeric variables. Hence, the QNP is a decidable fragment of numeric planning. The solution to a QNP problem is a policy that is a mapping from states to actions, where loop structures occur implicitly in a policy. Considering the integer numeric planning, GLINP is a higher expressive numeric planning framework than QNP. We find that many integer numeric planning domains, including Arith, Corner-A, D-Return, Hall-A, Prize-A and Sailing, can be formalized in GLINP, but cannot do in QNP since these domains require the formula to be in $\texttt{LIA}^\texttt{P}$ and the effect of actions to be accurate. Hence, GLINP has better applicability and wider scope than QNP.

Planning for possibly infinitely many initial states was firstly proposed in (Levesque 2005). Levesque (2005) developed a generation-and-test method to construct a planning program that solves problems for infinitely many states with a single numeric variable. This method consists of two stages: (1) search a plan without loop structure that works for the case where the value of the variable is less than a small threshold, and then try to roll the plan into a planning program; and (2) check if this solution solves all of large cases where the value is less than a large bound. Levesque (2005) also prove that for a very restricted domain, namely simple problems, there is a bound such that if the plan is valid for the value of the variable less than the bound, then it also holds for any value. The major shortcoming of the above method lies in the generation step which separates searching a plan without loop and rolling a program. In some cases, the size of plan is very large and hence the searching process takes prohibitively long time.

To overcome this shortcoming, Hu and Levesque (2009) used a different form of solutions, that is FSA (finite state automaton), instead of planning programs, and designed a method to generate FSA in the same way as Levesque (2005). The FSA is more general than the program, and a cycle in FSA can be considered as a loop structure. In the generation stage, Hu and Levesque (2009) constructs an FSA plan via constantly expanding new states and merging equivalent states according to the preconditions and effects of various actions until the FSA plan is valid for the instance with small threshold, or there does not exist such an FSA plan. The experimental results show that the method proposed in (Hu and Levesque 2009) scales better than that in (Levesque 2005). However, Hu and Levesque (2009) does not analyze the domains on which their method succeeds in all of the instances. To answer this question, (Hu and Levesque 2010) identified a class of generalized planning problems, namely one-dimensional (1d), that can be reduced to a planning problem with only one variable which is decremented by some actions. 1d problems are a broader class of planning problems than simple prolems. Generalized planning in 1d problems is decidable (Hu and Levesque 2010), and more precisely, in EXPSPACE (Hu and Giacomo 2011). The major advantage of our approach over the above methods is that our approach is able to solve the generalized planning problem with more than one numeric variable.

Srivastava, Immerman, and Zilberstein (2011) proposed a method to generate an FSA plan based on state abstraction using 3-valued logic (Sagiv, Reps, and Wilhelm 2002). It starts from an extended sequential plan $\pi \;:\; [(s_0, a_0), (s_1, a_1), \cdots, (s_n, a_n), (s_{n+1})]$ that is a sequence of state-action pairs $(s_i, a_i)$ with a final state $s_{n+1}$ entailing the goal. Then, it generates an abstract sequential plan $\pi' : [(S_0, a_0), (S_1, a_1), \cdots, (S_n, a_n), (S_{n+1})]$ that is obtained from the above concrete plan by generalizing each concrete state $s_i$ into an abstract state $S_i$ that covers a set of concrete states. Finally, based on the abstract sequential plan, it creates an FSA plan with cycle. If it detects two pairs $(S_j, a_j)$ and $(S_k, a_k)$ of $\pi'$ where $j < k$ are identical, then such repeated pairs means some properties that are true in $S_j$ after hold again in the successor abstract state $S_k$. Hence, a cycle representing the repetition of the sequence of actions $[a_j, a_{j+1}, \cdots, a_{k-1}]$ should be created in an FSA plan. However, the FSA plan generated by the above method has three drawbacks: (1) it is simple-loop; and (2) it does not guarantee goal achievement; and (3) the quality is sensitive to the extended sequential plan. In order to measure the condition when the plan is guaranteed to terminate and lead to the goal, namely *applicability condition*, Srivastava, Immerman, and Zilberstein (2012) designed an approach for computing such condition of the FSA plan when the plan is simple-loop. To extend the applicability condition of an FSA plan, Srivastava, Immerman, and Zilberstein (2010) provided a method to merging the sequential plan with an initial state $s_0'$ into the current FSA plan where the latter is not a solution for $s_0'$. However, the above works does not analyze the termination and executability properties of FSA plans. Later, an incremental approach for generating an FSA plan, which is terminable, executable and goal-reachable for all initial states, is developed by Srivastava et al. (2011a). The main idea of the approach is to iteratively extend a terminable and executable FSA plan $G$ according to new sequential plan for a state $s$ to which $G$ is not goal-reachable in $s$ until $G$ is a solution goal-reachable for all initial states. The difference between this approach and ours are as follows. Firstly, the solution to generalized planning is FSA plans while our solution is planning programs. Secondly, one of the advantages of our approach over theirs is that planning programs we synthesize can contain nested loop while the generation method proposed by Srivastava et al. (2011a) does not permit FSA plans with nested loop.

## Conclusions and Futrue Work

In this paper, we have introduced a scheme for computing generalized plan of numeric planning which solves planning problems for possibly infinitely many initial states rather than a single state. We summarize our main contributions as follows: Firstly, we propose a generalized version of numeric planning (GLINP), which is a more expressive planning formalization than QNP. Secondly, we propose a more expressive solution, namely planning programs, consisting of an empty plan, primitive actions, sequential, branch and loop structures. Thirdly, we develop a generation approach to synthesize the generalized plan. The generation

procedure synthesizes a skeleton of planning programs based on the idea of regex infering, and then completes the conditions of branch and loop structures by the enumerative algorithm. In theory, our algorithm (Alg. 2) for synthesizing skeletons of planning programs is proven to be sound, and our algorithm for completion of planning program (Alg. 4) is proven to be sound. Finally, we have implemented our approach and experimental results show the effectiveness and scalability of our proposed approach.

Our approach have a limitations, and hence leading to an avenues for future work. In this paper, the correctness verification is done manually, and the correctness of the planning program largely depends on the quality of the set of initial states. We would like to develop a generation-and-test approach to synthesize planning programs. The generation stage firstly infers a skeleton of planning programs, and then completes the conditions of branch and loop structures. The test stage verifies if the planning program is valid for a given GLINP problem. If it is correct, the expected planning program is found; otherwise, the generation stage restarts and constructs a new candidate. The whole process repeats until it finds a valid solution. To gurantee the correctness of planning programs, we verify whether programs satisfies following three critical properties: reachability, executability and termination.

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
