# OpenReview forum: "Generalized Linear Integer Numeric Planning"
_icaps-conference.org/ICAPS/2021/Workshop/HSDIP — Reject_

### Official Review · AnonReviewer2 · 2021-05-27
**.**

**Confidence:** 3
**Overall Score:** Reject

**Review:**

The paper proposes a method for solving generalized linear integer number planning
problems, where the goal is to find a program that is able to generate plans
for a (potentially infinite) set of planning tasks differing in the initial
state (defined as a formulae over numerical variables).

The idea of the paper is interesting, but the presentation needs a serious
revision, in my opinion. From the section "The Main Algorithm" onward, I
wasn't able to follow the explanations. The lines referred in the text are not
present in the pseudo-algorithms. So, it seems the text explanations were
written for a different version of the paper. The explanations are very
technical and convoluted, and mostly missing any high-level insight into the
proposed algorithms. For this reason, I think the paper should be rejected.


Minor issues:

Preliminaries:

In definition of Term: F is never defined. I assume it is a
function symbol, but is should be explicitly stated.

Second paragraph: "v \in \cal{F}" -- v is never used.

Last paragraph in left column: "Let \cal{V} a finite set ..." -> Let \cal{V}
denote a finite set ...

In the following definition of Term^P: F is, again, never defined, and v is
never used (but it is stated that v \in \cal{V})

page 2, last paragraph in right column: e^s is not defined

---

### Official Review · AnonReviewer1 · 2021-05-27

**Confidence:** 3
**Overall Score:** Weak Accept

**Review:**


Summary:

This paper introduces an algorithm for generalized numeric planning by producing a program that can solve multiple instances of the same domain.

The algorithm has the following steps:

(1) Generate some initial states and find a plan for each of them using a numeric planner
(2) Find a regular expression (RE) that describes all those plans
(3) Transform the RE into some planning program by adding conditions into the branching and loop decisions
(4) Validate that the program has generalized to unseen instances

In this paper, steps (1) and (4) are done manually.

Steps (2) and (3) could be trivial if done naively. Namely one could write a program that
solves all the input states by writing if s == s1 then apply plan1, else if s == s2 then
apply plan2, and so on. But this only works for the input states and not others. So the
key is to write a compact program that generalizes to unseen states.

To achieve a more compact program, the authors propose a method for generating regular
expressions that identifies common patterns and repetitions on the plans found, as well as
using a general method to find conditions on when to apply each plan.  While the proposed
method does not have any guarantees (e.g. on being the simplest program), it seems to
provide programs that generalize well across a number of domains.


In summary, I see this as a borderline paper for HSDIP. While it discusses interesting algorithmic ideas, the clarity should be greatly improved prior to publication. I am assigning it a weak accept score, but this would be conditional to improving the write-up in the final version if other reviewers agree that this is an option.




******************* Clarity ************************

The write up must be improved prior to publication, as there are a lot of unclear things in the description of the algorithm. I list some of the main difficulties I had below.

Part of the problem is that the The description in the text at the end of page 3 does not match at all the pseudocode. The text is describing more in detail steps (1) and (4), but they seem to have been removed from the paper (in the discussion they are mentioned as future work). Also, the Secttion is called "Generation of Initial States..." suggesting that step (1) will be discussed in detail. However, this part seems to have been partially removed, making the paper inintegible.

The paper would benefit a lot of having a running example. The sailing domain is described in the introduction, but then it is not used to illustrate the complex notions defined during the paper.
  - What would be the generalized version of sailing (used in the experiments)?
  - How the solution program looks like in the end?
  - What would be the intermediate regular expression skeleton?

In particular, the procedure for extracting regular expressions from plans is not entirely explained. Having some examples would be really helpful here. For example, what happens if we have a plan abab and another abc? do we reduce this to (ab)*(c|), or do we reduce it to ab(c|ab)? It seems that the current algorithm gives preference to repetitive structures, but some examples should be discussed. Reading the current paper it looks like there is no choice on what regular expression to use to represent the set of plans. But this is absolutely key for generalization. The current algorithm do not seem to consider any alternatives, so some explanation is needed to explain why the choices made by the algorithm are expected to work in practice.


The lines references from the text do not match at all the pseudocodes, please use label and ref in Latex in order to make sure that all the references point out to the right line number.

Algorithm 1:
 - The pseudocode says that a bound b is initialized. However, it is not clear what this bound is, or how it is initialized.
 -

Algorithm 2:
 - Line 4, should it be delta and not delta*?


Algorithm 3:
  - Line 13 has some ??
  - This algorithm is barely explained in the text and I could not follow the logic behind the pseudocode.
  - What is the purpose of l, why some recursive calls are necessary increasing it?
  - How the extension of the set delta exactly works (again, an example where this happens would be very helpful).

Algorithm 4:
 - I assume that Enumerate is calling to the algorithm by Udupa et al. but this is not well explained


In the experiments it is mentioned that you use Z3. What do you use Z3 for? (I saw that this is used by the algorithm in Udupa et al., but this should be clarified in the paper).

*******************  Evaluation ******************

The evaluation is slightly preliminar for a main conference, but I think it is sufficient for HSDIP. Nevertheless, if the authors plan to submit their work to a main conference in the future, I recommend the following:
 - The evaluation is not really showing the scalability of your approach, as you are evaluating only on very small problems. To test the scalability, you should increase problem size, evaluating how the runtime of your tool increases with respect to the size of the planning task, as well as the number of initial states considered.
 - It would be good to explain the plans found by the tool, perhaps in the form of supplementary material. As is, we cannot really judge the quality of the plans found. As I mentioned at the beginning of the review, finding a program that works for a finite set of states is trivial, and what is really interesting is to see how compact those programs are. In the text, it is mentioned that the plans have been manually validated, but this is really opaque for reviewers and/or other readers of the paper as we cannot verify this by ourselves

*******************  Related Work *******************

The paper discusses some related work. However, I still missed the discussion of relevant work for generalized planning.

In particular, the work by Javier Segovia Aguas is very closely related in that it derives generalized plans as planning programs, though in the simpler classical planning setting.
  - Javier Segovia Aguas, Sergio Jiménez, Anders Jonsson: Generalized Planning with Positive and Negative Examples. AAAI 2020: 9949-9956
  - Javier Segovia Aguas, Sergio Jiménez Celorrio, Anders Jonsson: Computing programs for generalized planning using a classical planner. Artif. Intell. 272: 52-85 (2019)
  - Generalized Planning as Heuristic Search (in ICAPS'21)
 (there are other papers that could be relevant as well).



Minor comments:

page 2:
Let Z the -> you need to add "be" or "denote", e.g. "let Z be". Similar in other places.
we use |pi| -> , pi should be introduced before where you mention a string.




page 4:
In Algorithm 2, in the Input, Sigma appears twice
In Algorithm 2, The while true if else break could be simplified using a do while construct, I think


page 5:
In Theorem 1,  r is the regex accepting each plan -> This suggests that there is a unique regex with this property, you should use "a regex" instead.
iteratively generates -> iteratively generate
by induction on size -> I did not understand this
excepted formula -> what do you mean by excepted?

page 7:
the height of tree -> of a tree.
In each domain, ..., the definition of other domains -> this sentence should be rewritten
propositional varialbes have only true and false -> this is true for all propositional variables, isn't it?
skelton -> skeleton

page 8,
cannot do in -> cannot be done in
after hold -> rephrase this
that planning programs we -> that the planning programs we
can contain nested loop -> loops

---

### Decision · Program_Chairs · 2021-06-10

**Decision:**

Reject

**Comment:**

We regret to inform you that your paper has not been accepted for publication at HSDIP'21.

There has been an internal discussion among the reviewers and organizers. While there is an agreement that there is value in the ideas behind the paper, the write-up needs to be improved prior to publication by fixing most of the points indicated in the reviews.

In particular, the algorithms' pseudocode should be understandable from the text explanation and in the current version there is a mismatch between the two.

We hope that future iterations can further improve the paper clarity and we will welcome them at HSDIP.